# Protein Hydrolysate or Plant Extract-based Biostimulants Enhanced Yield and Quality Performances of Greenhouse Perennial Wall Rocket Grown in Different Seasons

**DOI:** 10.3390/plants8070208

**Published:** 2019-07-05

**Authors:** Gianluca Caruso, Stefania De Pascale, Eugenio Cozzolino, Maria Giordano, Christophe El-Nakhel, Antonio Cuciniello, Vincenzo Cenvinzo, Giuseppe Colla, Youssef Rouphael

**Affiliations:** 1Department of Agricultural Sciences, University of Naples Federico II, 80055 Portici (Naples), Italy; 2Council for Agricultural Research and Economics (CREA)—Research Center for Cereal and Industrial Crops, 81100 Caserta, Italy; 3Department of Agriculture and Forest Sciences, Tuscia University, 01100 Viterbo, Italy

**Keywords:** *Diplotaxis tenuifolia* (L.) DC., sustainable horticulture, natural biostimulants, production, functional and nutritional quality, nitrate

## Abstract

Research has been increasingly focusing on the environmentally friendly biostimulation of vegetable crop performances under sustainable farming management. An experiment was carried out in southern Italy on *Diplotaxis tenuifolia* to assess the effects of two plant biostimulants (Legume-derived protein hydrolysate, Trainer^®^; Tropical plant extract, Auxym^®^) and a non-treated control, in factorial combination with three crop cycles (autumn–winter; winter; and winter–spring) on leaf yield, photosynthetic and colour status, quality, elemental composition, antioxidant content and activity. Both biostimulants prevalently contain amino acids and soluble peptides, showing the major effects on crop performances, though Auxym also has a small percentage of phytohormones and vitamins. The biostimulants enhanced plant growth and the productivity of perennial wall rocket. The winter–spring cycle led to higher leaf yield than the winter one. The two plant biostimulants enhanced leaf dry matter, oxalic and citric acids, Ca and P concentrations, phenols and ascorbic acid content as well as antioxidant activity, but did not increase nitrate content. A presumed mechanism involved in the enhancement of crop production could be attributed to the improvement of mineral nutrient availability and uptake. The winter–spring cycle elicited higher antioxidant content and activity than winter crops. Our current study shows that both the legume-derived protein hydrolysate and tropical plant extract represent an effective tool for boosting the yield, nutritional and functional quality of vegetable produce in the view of sustainable crop systems.

## 1. Introduction

Perennial wall rocket (*Diplotaxis tenuifolia* (L.) DC.) is an important leafy vegetable crop mostly cultivated in greenhouses, with Italy being the European leader in the production of this species, addressed both to the fresh market and baby leaf industry, with an estimated surface area of about 4800 ha in 2018 [1]. The increasing diffusion of perennial wall rocket in the last two decades is due to its smooth and succulent leaves, which meet consumers’ expectations. The leaves of this species are also rich in mineral elements and antioxidants [2,3].

The vegetable system management is targeted to fulfil the increasing world population and market demand by enhancing the yield, nutritional and functional quality of produce and concurrently safeguarding the environment [4]. Plant biostimulants are substances and/or microorganisms representing an innovative environmentally friendly tool for valorising plant nutrition, strengthening the response to abiotic stressors, modulating the quality of edible plant parts [5], boosting and stabilizing yield [6,7,8,9,10]. These compounds are defined as “CE marked products which stimulate plant physiological processes independently on their nutrient content by improving one or more of the following characteristics of the plant rhizosphere or phyllosphere: (i) nutrient use efficiency; (ii) tolerance to abiotic stress; (iii) crop quality; (iv) availability of confined nutrients in the soil and rhizosphere” [11,12]. When sprayed onto leaves, they are absorbed through the cuticle, epidermal cells and stomata, and finally reach the mesophyll cells [13]. 

Within plant biostimulants, protein hydrolysates (PHs) are “a mix of free amino acids as well as oligo- and polypeptides derived by chemical, enzymatic or chemical-enzymatic hydrolysis of plant residues or animal tissues” [14]. In particular, the enzymatic hydrolysis of proteins is ecologically safe [15] and compatible with organic farming [16,17], though use efficiency and economic feasibility of the production techniques of this biostimulant type should be further improved [18]. 

In the last decade, protein hydrolysates and natural plant extracts including those of tropical origin have been widely used as plant biostimulants for their beneficial effects on crop productivity and nutritional efficiency [16]. Their biostimulant function in relation to root growth and leaf biomass in many plant species is connected to signaling compounds/molecules, in particular amino acids and peptides, and to a lesser extend carbohydrates and lignosulphonates [5]. These can be contained in plant-derived biostimulants such as protein hydrolysates or vegetal extract-based products [19], or can be generated after their application through microbial activity. Indeed, plant biostimulants or their degradation products can affect the activity of epiphytic microbes on plant growth [20,21]. In this respect, foliar application of natural plant biostimulants can promote the development of beneficial epiphytic bacteria in plants [22], which enhance plants’ ability to absorb nutrients and self-protect against abiotic stressors and suboptimal conditions [23,24,25]. 

The positive effects of plant biostimulants on yield and especially on functional quality have been reported on different fruit and leafy vegetable crops including tomato, pepper and spinach [8,26,27]. For instance, drench application of a commercial extract of brown macroalgae (*Ascophyllum nodosum*) at 1.0 g L^−1^ was found to stimulate flavonoid synthesis by boosting total antioxidant capacity as well as total phenolics in spinach [28]. Similarly, Ertani and co-workers [26] demonstrated that the application of 50 mL L^−1^ of alfalfa-derived PH incurred a significant increase in antioxidant activity and target phenolic acid (i.e., chlorogenic acid) in green pepper fruits. 

Indeed, the mentioned authors [8,26,27,28] associated the beneficial effects of plant biostimulants, in particular PH and plant-extract (PE), to several direct and indirect physiological and molecular mechanisms, such as (i) stimulation of key enzymatic activities that correlate with the N metabolism and the elicitation of target hormone-like activity (auxin and gibberellin; direct mechanism) and (ii) enhancing the nutritional status of plant biostimulant-treated plants through the modification of the root apparatus in terms of biomass, root density and lateral root branching which enhance macro- and micro-nutrients uptake, assimilation and translocation [7,29,30]. 

Taking into account the influence of crop system components on plant biostimulant action which is missing from the scientific literature, we conducted a study on greenhouse grown perennial wall rocket with the aim of assessing the interaction between two biostimulant formulates (Legume-derived protein hydrolysate, PH, or Tropical plant extract, PE) and three crop cycles (autumn–winter; winter; winter–spring) on yield, colorimetric parameters, mineral profile as well as the functional quality of perennial wall rocket.

## 2. Results and Discussion

### 2.1. Implications of Crop Cycle and Biostimulant Application on Plant Growth and Yield

The plant determination variables examined in our research were not significantly affected by year, therefore, we only report mean data of the two years. Moreover, we identified no significant interactions between the two experimental factors ‘crop cycle’ and ‘biostimulant’. Therefore, only the data relevant to their main effects are shown in Table 1, Table 2, Table 3, Table 4 and Table 5.

As reported in Table 1, the autumn–winter crop cycle was the longest (69 days) and the winter–spring was the shortest (33 days). The winter–spring cycle resulted in higher leaf area index (LAI) and plant dry matter compared to the winter crop, in addition to higher marketable yield (+14%) due to the higher mean leaf weight. However, the number of leaves was lower; the autumn–winter crops did not show significant differences in comparison to those grown in winter and winter–spring (Table 1). 

Biostimulant application did not affect the crop cycle length, but enhanced the leaf surface expansion and plant biomass accumulation compared to the non-treated control (Table 1). Both biostimulant formulates showed a better effect on marketable yield compared to control (+12%) as a consequence of the higher number of leaves whose mean weight was not significantly affected, however, they did not differ from each other. 

In the present research, the two plant biostimulants applied did not result in different effects, though PE contains a small percentage of phytohormones and vitamins, in addition to carbohydrates, amino acids and peptides (Figure 1). Indeed, the two latter nitrogen-based compounds represent the major components of both biostimulants (75% in PH and 54% in PE) and have presumably exerted the most significant action on the vegetative growth of a leafy species such as perennial wall rocket. In this respect, nitrogen encouraged plant growth, reflected by the higher values of leaf area expansion and dry matter accumulation, which are associated with the higher leaf yield elicited by the two biostimulant formulates compared to the untreated control. Notably, the short crop cycles of perennial wall rocket, which coincide with the plants’ vegetative phase, did not show the need for additional contribution from phytohormones and vitamins.

In previous research, commercial plant biostimulants based on tropical plant extract (PE) or vegetal-derived protein hydrolysate (PH) increased the leaf value of SPAD (Soil Plant Analysis Development) index and fresh plant biomass of lettuce by 25.0% and 10.6%, respectively, compared to the non-treated control [31]. Presumably, the amino acids, peptides and phytohormones contained in the mentioned formulates stimulated the biomass increase, also as a consequence of doubling the number of cultivable epiphytic bacteria and increasing the species’ richness and diversity indices compared to the non-treated plants [31]. Furthermore, the positive effect of PE and PH on plant growth and crop productivity could be attributed to: (i) the stimulation of cell proliferation associated with the presence of signalling molecules such as key amino acids (i.e., glutamic and aspartic acids involved in the N metabolism) and soluble peptides; (ii) the protection of plant cells from oxidative damage exerted by vitamins; and (iii) plant metabolism enhancement by micronutrients [22]. In this respect, Kulkarni et al. [32] reported an important increase in the contents of dihydrozeatin, ciszeatin and isopentenyladenine types of cytokinins in biostimulant-treated spinach plants compared to the untreated control. In addition to the former direct mode of action of plant biostimulants, the application of PE and legume-derived PH improved the uptake, and thus the assimilation, of macronutrients by modulating the root system architecture (expressed in terms of biomass, root density and length as well as higher number of lateral roots), and also enhancing microbial activity and accordingly increasing soil nutrient availability [5,30]. All the above-mentioned direct and indirect mechanisms may have boosted plant growth parameters as well as marketable yield in biostimulant-treated rocket compared to the control treatment. 

### 2.2. Implications of Crop Cycle and Biostimulant Application for SPAD index and Leaf Colorimetry

The SPAD index recorded on perennial wall rocket leaves was not significantly affected by the crop cycle (Table 2). Biostimulant application resulted in higher values of this indicator of plant photosynthetic status compared to the non-treated control. This result suggests that both biostimulant formulates promoted the increase of leaf chlorophyll content, as SPAD represents a non-destructive estimate of this soluble pigment.

In previous research, the leaf SPAD index showed a 38.9% average increase in lettuce leaves under the foliar application of biostimulants based on tropical plant extract or vegetal-derived protein hydrolysate [31], and a 18.9% increase in spinach leaves upon the legume-derived protein hydrolysate spray [11], compared to the non-treated control.

Presumably, the different compounds contained in the applied biostimulants enhanced N uptake efficiency, as shown by the increase in SPAD index values. Indeed, this index is deemed a major indicator of the efficiency of green pigment (i.e., chlorophyll) biosynthesis and photosynthetic apparatus contributing to photosynthate transport through the phloem from sources to sinks, thus improving crop outcome [22,33].

With regard to the colorimetric components based on CIELAB indications, which significantly orient consumer choices of vegetable produce [34], the L* indicator (brightness) was not significantly affected by the crop cycle and the biostimulant application (Table 2). The leaves produced both in the winter–spring season or under biostimulant application were greener, showing a lower a* (redness) value compared to those reared in the winter and non-treated control. Finally, the winter–spring crops also resulted in higher values of the b* component (yellowness), which was not significantly influenced by the biostimulant formulate.

### 2.3. Implications of Crop Cycle and Biostimulant Application for Leaf Quality and Mineral Composition

The leaves grown in the winter cycle showed higher values of dry residue, oxalic and isocitric acid compared to those reared in the winter–spring (Table 3). Biostimulant application enhanced the leaf dry residue as well as the content of oxalic and citric acids compared to the non-treated control, however, no significant differences were found between the two formulates tested.

In previous research [11], the protein hydrolysate-based biostimulant did not show significant effects on spinach leaf dry matter. In a recent study, Paul et al. [35] reported that foliar application of PH treatment reprogrammed the metabolic profile of tomato plants through complex signalling mechanism that involved the direct precursor of both ethylene and polyamine conjugates. Among plant growth regulators, the ethylene precursor 1-aminocyclopropane-1-carboxylate (ACC) accumulation was encouraged in biostimulant-treated plants, entailing an ethylene increase as well [35]. Notably, polyamine conjugates accumulate in plants upon biostimulant treatment and are implicated in the better crop performance under optimal and sub-optimal conditions [36,37,38,39]. 

It is well known that mineral elements provide an essential contribution to the human organism’s metabolism [40]. Among the mineral elements analysed in perennial wall rocket leaves in the present research, the content of potassium and sulphur was higher in winter–spring, whereas calcium and nitrate accumulated at higher levels in the winter (Table 4). Compared to the non-treated control, the biostimulant application led to a higher content of calcium and phosphorus, and no nitrate difference. Notably, nitrate showed the highest concentration (63.8 g·kg^−1^ d.w. on average), followed by potassium (52.9 g·kg^−1^ d.w,) and calcium (26.7 g·kg^−1^ d.w,), which was 23-fold higher than the lowest one recorded for phosphorus.

In previous experimental trials [8,10], tomato plants sprayed with a legume-derived protein hydrolysate showed higher K and Mg compared to the non-treated control. Colonna et al. [41] reported a prevalent content of K compared to other nutrients in greenhouse grown baby-leaf spinach. Furthermore, Rouphael et al. [11] found that the foliar treatment with a legume-derived protein hydrolysate resulted in 36.4% and 25.0% leaf increase of K and Mg contents, respectively, and a lower Na/K ratio (0.014 vs. 0.025) in the same vegetable species compared to non-treated plants. Indeed, the low Na/K ratio is likely to cause lower incidence of heart attacks and hypertension [42]. The effect of legume-derived protein hydrolysate on improving the nutritional status has also been recorded in greenhouse tomato fruits [8].

The increased content of K and Mg in perennial wall rocket leaves may have stemmed from: the action of signalling molecules (i.e., soluble peptides and key amino acids) [22,43]; the modified root architecture leading to enhanced nutrient uptake, translocation and accumulation [16,44]; the gene expression for macronutrient transporter encoded in cell membranes [33,45].

Perennial wall rocket plants show high leaf accumulation of nitrate, whose excess is potentially harmful to human health [46]. In the present research, the biostimulant application did not cause an increase in this ion’s concentration compared to the non-treated control (Table 4). However, nitrate concentration never exceeded the thresholds related to rocket leaves set by EC Regulation No 1258/2011 to 6000 mg·kg^−1^ f.w. (1 April to 30 September) or 7000 (1 October to 31 March). As for total nitrogen, we found the growth of perennial wall rocket plants to be enhanced by both biostimulants compared to the non-treated control (Table 1), and this assessment may witness the higher N assimilation elicited by PH and PE. Protein hydrolysate prevented nitrate accumulation in perennial wall rocket leaves as a consequence of gene up-regulation connected to nitrate reductase, which led to higher conversion of this ion into amino acids [47]. In previous research [45], protein hydrolysate was shown to modulate plant growth and the expression of key genes in N assimilation, including nitrate and ammonia transporters in tomato. Trevisan et al. [48] found an evident regulation of gene transcription in maize relevant to the high affinity nitrate transport system, which is also involved in nitrogen use efficiency. Vernieri et al. [49] recorded a leaf nitrate content reduction in rocket upon the application of the commercial biostimulant Activawe containing amino acids. Finally, nitrate concentration in leafy vegetables such as lettuce, pack choi, rocket, spinach and Swiss chard, was reduced by the application of protein hydrolysates as well as mixed or single amino acids [50,51,52]. 

### 2.4. Implications of Crop Cycle and Biostimulant Application for Antioxidant Compounds and Activity

Perennial wall rocket leaves grown in winter–spring showed higher content of total phenols and total ascorbic acid compared to winter ones, by 93% and 152%, respectively (Table 5). Consequently, the lipophilic and hydrophilic antioxidant activities were higher in the winter–spring season. The two biostimulant formulates applied to rocket plants resulted in higher antioxidant compounds and activity compared to the non-treated control, but they did not display different effects with respect to each other (Table 5).

Perennial wall rocket leaves are characterized by high levels of antioxidant compounds and activity [3], which are reportedly health-beneficial [53]. The synthesis and accumulation of antioxidant compounds may be connected both to the enzymatic activity involved in phytochemical homeostasis [10,26], and high tissue K and Mg content [10]. In previous research, protein hydrolysates enhanced ascorbate, p-coumaric, chlorogenic acid and capsaicin concentrations as well as antioxidant activity of *Capsicum chinensis* L. in greenhouse pepper fruits [26]; in addition to ascorbic acid content and antioxidant activity of greenhouse grown tomato fruits [8,10]. Kulkarni et al. [32] reported the positive effect of biostimulant application in increasing spinach content in phenylalanine ammonia lyase, an important enzyme [54] involved in the biosynthesis of phenolic acids. Similarly, Jȩdrszczyk et al. [55] recorded a higher antioxidant activity in garlic leaves upon humic acid application. Vasantharaja et al. [56] reported a positive biostimulant effect related to the application of 3% *Sargassum swartzii* extract on phenols and antioxidant activity of *Vigna unguiculata*. Furthermore, the protein hydrolysate and plant extract-based biostimulants primed maize plants to get protection against oxidative stresses by enhancing the expression of genes regulating the activity of superoxide dismutases [48], which are key enzymes related to the antioxidant defence by catalysing the enzymatic dismutation of superoxide to H_2_O_2_ [57].

## 3. Materials and Methods 

### 3.1. Plant Material and Growing Conditions 

Our research was carried out at University of Naples Federico II, Portici (Naples, southern Italy, 40° 49′ N, 14°15′ E; 72 m a.s.l.) in 2016–2017 and 2017–2018 growing seasons on perennial wall rocket (*Diplotaxis tenuifolia* (L.) DC.) cultivar ‘Nature’, in an unheated greenhouse made of three spans covered with a long-life thermal polyethylene film, with an overall width of 15.0 m, length 30.0 m and heights of 2.0 and 3.5 m at the wall and roof, respectively. The soil was sandy-loam (76%, 17% and 7%, sand, silt and clay, respectively), with pH 6.9, electrical conductivity = 512 mS·cm^−1^, organic matter = 2.25% (*w*/*w*), total nitrogen = 0.14 %, P = 32.8 mg kg^−1^ and exchangeable K = 1,372 mg kg^−1^. The trend of temperature is shown in Figure 2 as mean values of the two research years, because no variable regarding the plant determinations was affected by the year of investigation. 

The cultivar ‘Nature’ is widely diffused in the Campania region where the experiment was conducted. Sustainable crop management was performed, with the following pre-transplant practices: soil plough and hoeing at 20 cm depth; organic fertilization with 38 kg·ha^−1^ N, 10 P_2_O_5_ and 30 K_2_O; 100 cm wide raised beds mulched with a 15 µm thick MaterBi biodegradable black film. The transplant was performed on 16 and 20 November in 2016 and 2017, respectively, with 20 × 20 cm plant spacing within each bed, with 80 cm between the outer rows of adjacent beds (14.3 alveoli per m^2^). During the crop, the farming practices were: 112 kg·ha^−1^ N, 30 P_2_O_5_ and 90 K_2_O supply through fertigation; drip irrigation coinciding with 80% soil available water capacity at 20 cm depth; six foliar applications against fungal diseases and insects using copper (0.7 kg·ha^−1^ copper oxichloride) and azadirachtin (25 mL·ha^−1^ active ingredient). 

Harvests of commercially ripe leaves (12 to 15 cm length) were performed, practicing the cut at 3–5 cm above the cotyledons in order to allow for efficient vegetative apex regrowth [58,59], as follows: 24 and 27 January in 2017 and 2018, respectively, for the first crop cycle; on 6 and 8 March in 2017 and 2018, respectively, for the second crop cycle; on 9 and 10 April in 2017 and 2018 respectively, for the third crop cycle.

### 3.2. Experimental Protocol and Treatments Application 

A factorial combination between three crop cycles of perennial wall rocket (autumn–winter, winter, winter–spring) and two biostimulant formulates (legume-derived protein hydrolysate, Trainer; tropical plant extract, Auxym) plus a non-treated control were adopted in the present greenhouse experiment. Treatments were arranged in a randomized complete block design with three replicates, and the experimental unit had a 3.2 m^2^ surface area with 80 plants, of which, 36 were used for yield determinations.

The two commercial plant biostimulants, ‘Trainer’ and ‘Auxym’, were provided by Italpollina S.p.A., Rivoli Veronese, Italy. The vegetal-derived PH ‘Trainer’ was obtained through enzymatic hydrolysis of proteins derived from legume seeds. It contains mainly free amino acids and soluble peptides. ‘Trainer’ also contains the following macro-elements (g·kg^−1^): N = 50.0, P = 0.9, K = 4.1, Ca = 0.7 and Mg = 10.0; and micro-elements (mg·kg^−1^): Fe = 30.0, Mn = 1.0, B = 1.0, Zn = 9.6 and Cu = 9.0. The aminogram of ‘Trainer’ has been reported in detail by Colla et al. [8]. ‘Auxym’ is a biostimulant based on tropical plant extract (PE) obtained upon water fermentation. The tropical plant extract contains phytohormones (mainly auxin and cytokinins), amino acids and peptides, vitamins and micronutrients as reported previously by Rouphael et al. [10]. 

Perennial wall rocket plants were uniformly sprayed with a solution containing 3 mL·L^−1^ of the protein hydrolysate Trainer or 2 mL·L^−1^ of the vegetal extract Auxym, or just with water as a control treatment, three times during the growing cycle at seven-day intervals, starting when the leaves were 6 cm long. The perennial wall rocket plants were uniformly sprayed using a 16-L stainless steel sprayer ‘Vibi Sprayer’ (Volpi, Piadena, Italy). The two commercial plant biostimulants were applied at concentrations complying with the recommendations from both the manufacturers and previously published papers [8,10].

### 3.3. Yield and Growth Assessment

At each harvest, a fresh yield of perennial wall rocket leaves was assessed in all plots, excluding the border plants, and twelve-plant samples were used for determining leaf number and mean weight. At the end of each crop cycle, dry weights were assessed by drying the plants at 70 °C until constant weight was reached. Total leaf area was measured using an Li-Cor3000 area meter (Li-Cor, Lincoln, NE, USA). The leaf dry matter percentage was also calculated.

For the determination of organic acids, nitrate content and leaf mineral composition, dry biomass was used, whereas for total phenols and ascorbic acid contents as well as for antioxidant activity, leaf samples were randomly taken from each plot, frozen in liquid nitrogen and stored at −80 °C until chemical analysis.

### 3.4. SPAD and Leaf Colour Parameters

Immediately prior to harvesting, the soil plant analysis development (SPAD) index was determined on twenty undamaged rocket leaves per experimental treatment, by means of a portable Konica Minolta chlorophyll meter (model SPAD-502, Tokyo, Japan). The leaf colour parameters L* (lightness, from 0 to 100, i.e., black to white), a* and b* (chroma components from −60 to + 60, i.e., from green to red and from blue to yellow for ‘a’ and ‘b’ respectively) were assessed [3].

### 3.5. Analysis of Mineral Elements

The desiccated rocket leaf tissues were ground and used for macro-mineral and organic acids profile analysis as described in detail by Rouphael et al. [10]. Nitrate, phosphorus, potassium, calcium, sulphur and magnesium were separated and quantified by ion chromatography (ICS-3000, Dionex, Sunnyvale, CA, USA) coupled to a conductivity detector. An IonPac CG12A (4 × 250 mm, Dionex, Corporation) guard column and IonPac CS12A (4 × 250 mm, Dionex, Corporation) analytical column were used for the K, Ca and Mg analysis; for nitrate, P, S and organic acids (malic, oxalic, citric and isocitric) determination, an IonPac AG11-HC guard (4 × 50 mm) column and IonPac AS11-HC analytical column (4 × 250 mm) were adopted. The macro-mineral and organic acids profiles were expressed in g kg^−1^ d.w.

### 3.6. Analysis of Antioxidant Molecules: Total Phenols and Ascorbic Acid

Total phenols and total ascorbic acid contents were assessed by spectrophotometric detection using the Folin–Ciocalteu method [60] and Kampfenkel [61], respectively. The solution absorbance for total phenols and total ascorbic acid was measured at 765 and 525 nm, respectively. 

### 3.7. Analysis of Antioxidant Activity

Lipophilic antioxidant activity (LAA) was determined using a radical cation assay, extracting 200 mg of lyophilized material by methanol. Based on the study by Re et al. [62], the 2,2′-azinobis 3-ethylbenzothiazoline-6-sulfonic acid (ABTS) method was used to measure LAA. The hydrophilic antioxidant activity (HAA) was measured using the *N*,*N*-dimethyl-p-phenylenediamine (DMPD) method [63] by extracting 200 mg of lyophilized material in distilled water. A UV–Vis spectrophotometer was used to measure the absorbance reduction of the solutions at 734 and 505 nm wavelength to determine LAA and HA, respectively. 

### 3.8. Statistical Processing

The two-way analysis of variance and DMRT were used for processing the experimental data and performing the mean separations at the 0.05 probability level, respectively, using the SPSS software version 21. The angular transformation was applied to percentage data before processing. 

## 4. Conclusions

From research carried out in greenhouses in southern Italy, we found that the foliar applications of two types of natural plant biostimulants, i.e., legume-derived protein hydrolysate or tropical plant extract, enhanced yield, photosynthetic and colour status, quality attributes, the content of Ca, P, phenols and ascorbic acid, and antioxidant activity of perennial wall rocket (*Diplotaxis tenuifolia* (L.) DC.) leaves over autumn to spring crop cycles. The effects of the two biostimulant formulates never significantly differed from each other, and neither elicited leaf nitrate content increase compared to the non-treated control. However, all the experimental treatments showed lower nitrate values compared to the thresholds set in the EC Regulation No 1258/2011. 

Overall, our results demonstrate that both plant biostimulants were able to trigger several physiological mechanisms, mainly based on the contribution of amino acids and peptides, which represent the major components of the two formulates applied. These nitrogen-containing compounds were shown to be essential for enhancing the growth of perennial wall rocket crops, coinciding with the vegetative phase. However, phytohormones and vitamins did not play a significant role, even during the short cycles of *D. tenuifolia*. Both PH and PE stimulated the synthesis and accumulation of important phytochemicals including ascorbate and phenols, which play a crucial role in boosting plant growth and at the same time constitute an additional value to the health of the human organism.

Based on the present investigation outcomes, growers can use plant biostimulants as a sustainable farming practice within the greenhouse leafy vegetable systems, in order to achieve yield increase and meet consumer expectations for premium-quality produce.

## Figures and Tables

**Figure 1 plants-08-00208-f001:**
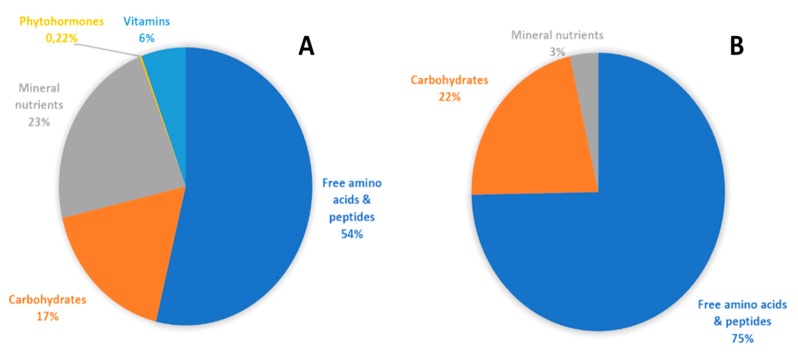
Main components (as percentage of the total) of tropical plant extract enriched with micronutrients (**A**) and vegetal-derived protein hydrolysate (**B**) tested in the present trial.

**Figure 2 plants-08-00208-f002:**
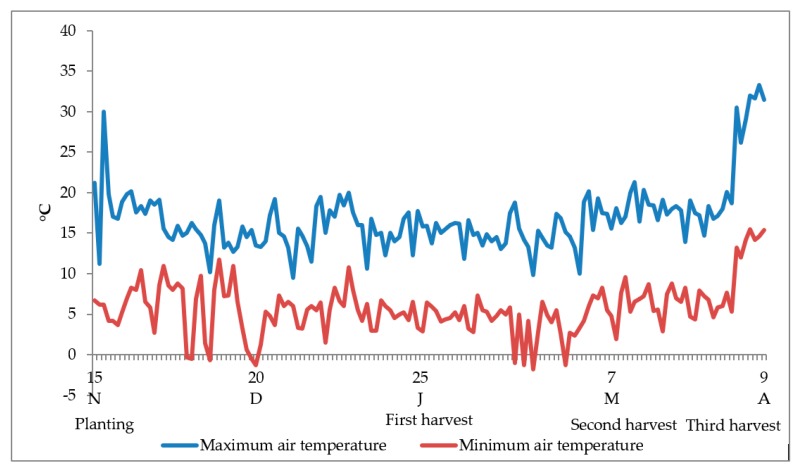
Trend of air temperature inside the greenhouse in Portici (Naples, southern Italy) as an average of 2016–2017 and 2017–2018.

**Table 1 plants-08-00208-t001:** Mean values of perennial wall rocket precocity, growth indices and yield components as affected by crop cycle and biostimulant.

Source of Variance	Crop Cycle Duration (Days)		Leaf Area Index (LAI) (m^2^·m^−2^)		Plant Dry Matter (g·m^−2^)		Marketable Leaves					
							Yield (t·ha^−1^)		Number per Alveolus		Mean Weight (g)	
Crop cycle												
Autumn-winter	69	a	1.40	ab	112.2	ab	12.4	ab	137.5	b	0.63	b
Winter	41	b	1.35	b	106.4	b	11.5	b	152.0	a	0.53	c
Winter-Spring	33	c	1.44	a	116.7	a	13.1	a	118.6	c	0.77	a
Biostimulant												
Non-treated	49		1.29	b	98.5	b	11.4	b	126.8	b	0.63	
Tropical plant extract (PE)	47		1.45	a	119.0	a	12.7	a	140.0	a	0.64	
Legume-derived protein hydrolysate (PH)	47		1.47	a	117.8	a	12.9	a	141.2	a	0.65	
	n.s.										n.s.	

Within each column: n.s., no statistically significant difference; means followed by different letters are significantly different according to the Duncan test at *p* ≤ 0.05.

**Table 2 plants-08-00208-t002:** Mean values of SPAD (Soil Plant Analysis Development) index and colour components of perennial wall rocket as affected by crop cycle and biostimulant.

Source of Variance	SPAD		L*	a*	b*
Crop cycle					
Winter	36.9		39.0	−12.9	18.9
Winter–Spring	38.8		40.3	−14.3	21.1
	n.s.		n.s.	*	*
Biostimulant					
Non-treated	35.8	b	38.6	−13.4	19.7
Tropical plant extract (PE)	38.4	a	40.0	−13.6	20.1
Legume-derived protein hydrolysate (PH)	39.3	a	40.3	−13.7	20.3
			n.s.	n.s.	n.s.

Within each column: n.s., no statistically significant difference; * significant difference at *p* ≤ 0.05; means followed by different letters are significantly different according to the Duncan test at *p* ≤ 0.05.

**Table 3 plants-08-00208-t003:** Mean values of perennial wall rocket leaf quality indicators as affected by crop cycle and biostimulant.

Source of Variance	Dry Matter		Organic Acids					
			Malic	Oxalic		Citric		Isocitric
	%			g·kg^−1^ d.w.				
Crop cycle								
Winter	9.25		26.4	0.88		21.0		0.64
Winter–Spring	8.84		25.9	0.80		21.4		0.58
	n.s.		n.s.	*		n.s.		*
Biostimulant								
Non-treated	8.54	b	25.5	0.78	b	19.7	b	0.60
Tropical plant extract (PE)	9.37	a	26.6	0.88	a	21.8	a	0.63
Legume-derived protein hydrolysate (PH)	9.23	a	26.3	0.87	a	22.1	a	0.61
			n.s.					n.s.

d.w., dry weight. Within each column: n.s., no statistically significant difference; * significant difference at *p* ≤ 0.05; means followed by different letters are significantly different according to the Duncan test at *p* ≤ 0.05.

**Table 4 plants-08-00208-t004:** Mean values of mineral composition of perennial wall rocket leaves as affected by crop cycle and biostimulant.

Source of variance	NO_3_	P		K	S	Ca		Mg
	mg·kg^−1^ f.w.	g·kg^−1^ d.w.						
Crop cycle								
Winter	6300	2.74		50.8	7.91	27.9		3.51
Winter–Spring	5260	2.68		55.0	8.84	25.5		3.20
	*	n.s.		*	*	*		*
Biostimulant								
Non-treated	5240	2.52	b	52.7	8.52	25.2	b	3.44
Tropical plant extract (PE)	5990	2.78	a	52.3	8.37	27.4	a	3.31
Legume-derived protein hydrolysate (PH)	6100	2.82	a	53.8	8.22	27.6	a	3.31
	n.s.			n.s.	n.s.			n.s.

f.w., fresh weight; d.w., dry weight. Within each column: n.s., no statistically significant difference; * significant difference at *p* ≤ 0.05; means followed by different letters are significantly different according to the Duncan test at *p* ≤ 0.05.

**Table 5 plants-08-00208-t005:** Mean values of antioxidant content and activity of perennial wall rocket leaves as affected by crop cycle and biostimulant.

Source of Variance	Polyphenols mg Gallic Acid 100 g^−1^ d.w.		Ascorbic Acid mg·100 g^−1^ f.w.		Lipophilic Antioxidant Activity mmol Trolox eq 100 g^−1^ d.w.		Hydrophilic Antioxidant Activity mmol Ascorbic Acid eq 100 g^−1^ d.w.	
Crop cycle								
Winter	206		23.1		9.27		6.45	
Winter–Spring	398		58.3		19.62		8.13	
	*		*		*		*	
Biostimulant								
Non-treated	278	b	25.6	b	11.53	b	6.61	b
Tropical plant extract (PE)	320	a	49.5	a	16.32	a	7.45	a
Legume-derived protein hydrolysate (PH)	308	a	47.0	a	15.50	a	7.80	a

f.w., fresh weight; d.w., dry weight. Within each column: * significant difference at *p* ≤ 0.05; means followed by different letters are significantly different according to the Duncan test at *p* ≤ 0.05.

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
