# Peer review of "Protein Hydrolysate or Plant Extract-based Biostimulants Enhanced Yield and Quality Performances of Greenhouse Perennial Wall Rocket Grown in Different Seasons"

_plants, 2019, doi:10.3390/plants8070208_

Round 1

Reviewer 1 Report

I am glad to chose a biostimulators. This is the future of vegetables production. Research is interesting and useful in applying horticulture to science.

Author Response

Dear Reviewer, thanks for your revision of our manuscript!

Reviewer 2 Report

I read with interest of MS "Natural Plant Biostimulants Enhance Yield and Quality Performances of Greenhouse Perennial Wall Rocket Grown in Different Seasons" The authors studied the effect of Legume-derived protein hydrolysate or Tropical plant extract development and three crop cycles on yield, colorimetric parameters, mineral profile as well as on functional quality of perennial wall rocket.
The results proposed are in part new. Nevertheless, due to the very complex effect of biostimulants that has yet to be untangled, the MS is interesting. The work is very interesting because of its practical significance.

The MS, however, presents several lacks.

In M&M (3.2. Experimental Protocol and Treatments Application):
Was the control combination sprayed with clean water? This information should be provided in MS.
Please indicate the operating pressure of the spray and the type of nozzle because the spraying technique can affect the tested characteristics.

Author Response

(The authors gave the same response as above.)

Reviewer 3 Report

This work is interesting because it is the first time that the effect of biostimulants is studied in relation to the crop cycle. The authors have characterized in a rather in-depth way two products with biostimulant activity, although the data are reported in a hurried way by making reference to previous studies. The results of the agronomic test are new and of interest. Nevertheless, it seems to me that the authors have been too quick to discuss the results obtained and have not spent some space trying to give a cause-effect explanation. Can be similarly stimulated physiological parameters by different molecules? Perhaps the end results are similar but achieved by activating different mechanisms. I believe this part needs to be developed. I think that this MS can be accepted after a thorough revision.

Title.

The title is too generic, especially regarding the first part. The term biostimulant, in itself, is a product of natural origin and not of synthesis. The term natural is not necessary. Why not contrast the two biostimulants in the title?

ABS

Add in the ABS the main chemical differences between the two biostimulants.

Line 27. It is not necessary to mention the SPAD index here.

The abstract should be partially rewritten considering an explanation regarding the mechanism of action-effect.

Introduction

Well done.

Results and discussion

In general: The authors should try to go further in defining the mechanism of action of the two biostimulants. If the two biostimulants differ in composition, is it possible that different molecules set in motion similar responses in plants? The authors should add a paragraph in this sense giving explanations or hypotheses.

Lines 110-115 and elsewhere. You shortened the term protein hydrolyzate with PH, I think you should also shorten the term tropical extract.

Lines 174-180. The nitrate content alone is not enough. The nitrate may be low, but if the total nitrogen is high it means that the nitrogen has been assimilated into proteins. The authors should provide the total nitrogen content (N protein) too.

Lines 240-243. Sentence relevant to the conclusions. Delete it from here or move it to the conclusion section.

Conclusions. As mentioned above, the authors should try to go further in defining the mechanism of action of the two biostimulants. If the two biostimulants differ in composition, is it possible that different principles set in motion similar responses in plants?

M&M

Lines 278-282. In the experimental design the actual number of plants used is not stated. Add it.

Line 290. Tropical plant: Is it not possible to indicate the type of plants used?

Lines 285-291. It is good that the characterizations of the two biostimulants have already been published in previous works, nevertheless the MS would be more complete, and more appreciable, if the authors introduce these data perhaps in a graphic form (e.g. as circular diagram or pie chart) or as ratios. The authors must also add one or two sentences to emphasize the differences and/or similarities between the two products.

Lines 295-296. Generic. Please, provide the concentrations used.

Lines 345-349. Sentence relevant to the results. Move it to the results section.

Author Response

(The authors gave the same response as above.)

Round 2

Reviewer 3 Report

The authors did a good job and answered what was requested. The manuscript, in my opinion, is ready to be published.